# Short-term effects of side-alternating Whole-Body Vibration on cognitive function of young adults

Y. Laurisa Arenales Arauz[1,2], Eddy A. van der Zee[3], Ype P. T. Kamsma[1], Marieke J. G. van Heuvelen[1]*

**1** Department of Human Movement Sciences, University of Groningen, University Medical Center Groningen, Groningen, The Netherlands, **2** Human Physiology and Sports Physiotherapy Research Group, Vrije Universiteit Brussel, Brussels, Belgium, **3** Molecular Neurobiology, Groningen Institute for Evolutionary Life Sciences (GELIFES), University of Groningen, Groningen, The Netherlands

* m.j.g.van.heuvelen@umcg.nl

**Data Availability Statement:** All relevant data are in the Supporting information file.

**Funding:** The author(s) received no specific funding for this work.

## Abstract

Recent research in rodents and humans revealed that Whole-Body Vibration (WBV) is beneficial for cognitive functions. However, the optimal WBV conditions are not established: contrary to vertical WBV, side-alternating WBV was not investigated before. The present study investigated the short-term effects of side-alternating WBV in standing and sitting posture on specific cognitive function of young adults. We used a balanced cross-over design. Sixty healthy young adults (mean age 21.7 ± 2.0 years, 72% female) participated. They were exposed to three bouts of two-minute side-alternating WBV (frequency 27 Hz) and three control conditions in two different sessions. In one session a sitting posture was used and in the other session a standing (semi-squat) posture. After each condition selective attention and inhibition was measured with the incongruent condition of the Stroop Color-Word Interference Test. WBV significantly ($p = 0.026$) improved selective attention and inhibition in the sitting posture, but not in the standing posture. The sitting posture was perceived as more comfortable, joyous and less exhaustive as compared to the standing posture. This study demonstrated that side-alternating WBV in sitting posture improves selective attention and inhibition in healthy young adults. This indicates that posture moderates the cognitive effect of WBV, although the effects are still small. Future studies should focus on the working mechanisms and further optimization of settings, especially in individuals who are unable to perform active exercise.

## Introduction

Regular exercise is essential for healthy ageing and helps to maintain cognitive performance [1, 2]. Due to age-related disabilities and/or lack of supervision, it can be challenging to engage in regular exercise. Therefore, alternative and more feasible methods of exercise are needed. In this context, Whole-Body Vibration (WBV) is receiving increased attention [3–6].

**Competing interests:** The authors have declared that no competing interests exist.

WBV is an exercise modality or therapeutic method in which subjects are exposed to mechanical oscillations through a vibrating platform. The vibrations are transmitted through the body while maintaining a static (standing, sitting or lying down) or active (dynamic exercises) posture. The types of vibration transmissions can be categorized into vertical vibration transmission and side-alternating vibration transmission [7]. The intensity of WBV is further defined by frequency, peak-to-peak displacement and temporal aspects such as number and durations of bouts. There is ample evidence for positive effects of WBV on physiological measures such as muscle strength [8–10], balance [11, 12], mobility [13], bone mineral density [14–17], blood flow [18], oxygen uptake [19] and heart rate [19], however, the impact of WBV on cognition and the brain is less extensively explored.

Several animal studies support a positive impact of WBV on cognition and the brain [20–22]. After WBV training, mice showed improvements in memory and learning performance using novel object recognition [21]. In aged rats, spatial memory improved by WBV training whereas object memory did not change [20]. Although the exact underlying mechanisms of these effects are currently unknown, several potential mechanisms are identified. First, WBV stimulates skin mechanoreceptors (Meissner and Pacinian corpuscles) specifically sensitive to vibrations of 30-40Hz [23]. By stimulation of these corpuscles, afferent signals are transmitted via the spinal cord and thalamus to the primary somatic sensory cortex. From these areas, direct and indirect connections to the prefrontal cortex and other brain regions induce enhanced neurotransmission [24]. Indeed, a functional near-infrared spectroscopy (fNIRS) study in humans provided evidence for increased cortical activity in motor networks and prefrontal cortical areas during WBV training [25]. Other animal research revealed that daily sessions of WBV for a period of five weeks increased activity of the cholinergic system in the somatosensory cortex and amygdala, which is considered crucial for learning and memory performance [26]. WBV further increased the expression of c-fos protein, a brain marker for neuronal activity in brain regions specific to learning and memory functions [24]. Other mechanisms include an increase in synaptic plasticity [27], neurogenesis [28, 29], neurotransmitters [30, 31], Brain Derived Neurotrophic Factor (BDNF) [31, 32] and Glucose Transporter 1 (GLUT1) immunoreactivity [33] in specific regions in the animal brain after WBV intervention.

Additional to these brain mechanisms, it is demonstrated that WBV activates muscle sensory receptors (muscle spindles), which repeatedly initiate reflex muscle contractions while standing on the platform [34]. The constant initiation of muscle contractions induced by WBV causes a secretion of myokines in the bloodstream which might contribute either directly or via enhanced BDNF towards improved cognition [35]. WBV further reduces motor recruitment thresholds [36] leading to enhanced neuromuscular function and reciprocal inhibition [37].

In our research group, we performed several studies in humans on the short-term and chronic effects of vertical WBV while sitting on a chair mounted on a WBV device [38–42]. We consistently found significant short-term effects in favor of WBV vs. control condition on selective attention and inhibition in young healthy adults [39, 41], 8-13-year-old children [40] and young adults with Attention Deficit Hyperactivity Disorder (ADHD) [39]. Similarly, we found significant improvements in selective attention and inhibition after a five-week intervention in older adults [42] and after ten days in a subject with ADHD [38]. However, although significant effects were obtained, effects sizes were generally small to moderate and the clinical relevance of the improvements can be considered controversial. Furthermore, we did not find significant improvements in psychomotor speed [41], reaction time [38], memory and cognitive flexibility [42]. Research to identify optimal settings is warranted. In this perspective, another type of vibration transmission (side-alternating instead of vertical) might

bring more insights. Therefore, we focus on the effects of side-alternating WBV on cognition in the present study.

Side-alternating WBV generates vibrations along the sagittal axis causing a tilting movement of the platform. Compared to vertical vibration transmission, side alternating vibrations lead to higher neuromuscular activity [43, 44], heart rate response and oxygen consumption [19]. Despite using similar input vibration characteristics, side-alternating WBV also produced greater acceleration magnitudes through the lower body [45–47]. If an increase in physiological response contributes to improved cognitive performance, side-alternating vs. vertical WBV might be more effective. However, as far as we know, the cognitive effects of side-alternating WBV are not investigated yet.

The posture or body position on the WBV platform is considered a key parameter for the transmission of vibration through the body [48]. Our previous studies on the impact of WBV on cognition, applied vertical WBV while participants sat on a chair mounted on the vibrating platform. Evidence related to the cognitive enhancing effect of WBV in a standing posture is limited. Two studies provided positive acute effects of vertical WBV in a standing posture on reaction time [49], cognitive function and brain activation measured with Electroencephalography (EEG) [50] in fragile populations aged 65 years and older. Active exercises with vertical WBV revealed conflicting results. Cognitive function and cerebral blood flow increased [51] whereas cognitive status remained unchanged in elderly women [52]. In a standing posture, the constant feedback mechanisms of the central equilibrium system and activation of stretch reflexes may attribute to the elevated cognitive performance [24, 34]. The sitting posture might result in a higher activation of skin mechanoreceptors contributing to an increase in cognitive performance. Therefore, in the present study, we examine both sitting and standing postures on the WBV device.

The main objective of this study is to investigate the short-term effects of side-alternating WBV on selective attention and inhibition in sitting and standing posture in young adults. Based on the results of previous studies, we expect that subjects will increase cognitive performance after WBV training. A possible moderating effect of posture may give more insight into the underlying mechanisms of WBV affecting cognition.

## Methods

We used the general reporting guidelines of the Consolidated Standards of Reporting Trials (CONSORT) statement, which is an evidence-based, minimum set of recommendations for reporting randomized trials [53]. In addition, we used the specific reporting guidelines for WBV studies in humans [7].

### Study design

A cross-over randomized controlled trial was conducted. All subjects underwent alternating three WBV and three control conditions in both sitting and standing position. The order of condition (WBV or control) and posture (sitting or standing) were balanced whereby the starting condition and starting position were randomly assigned. Randomization of the conditions and postures was done by an independent researcher using random numbers for each gender separately with an allocation ratio of 1:1:1:1.

### Ethics

The study protocol was approved by the local ethical committee of the department of Human Movement Sciences of the University Medical Center Groningen and is registered in the

UMCG register with registration number 2020000035. We obtained written informed consent from all participants prior to the procedure.

## Participants

Sixty healthy students were recruited from the University of Groningen, the Netherlands (see Table 1 for participant characteristics). The study was carried out in two research labs located in the faculty of Medical Science and the faculty of Science and Engineering between February and October 2021. Participants did not receive any financial or other compensations for their contribution. However, participation could be rewarded with research requirement hours for undergraduate students. To ensure safety, the Physical Activity Readiness Questionnaire (PARQ; [54]) was assessed before participation. Inclusion criteria were 1) educated at university level 2) aged between 18–30 years, 3) meeting the criteria of the PARQ. Exclusion criteria were 1) red-green color deficiency, 2) history of developmental or psychiatric disorders.

## Materials

**Vibration device.** WBV was applied using the WBV Galileo Fit Sensor platform (manufactured by Novotec Medical GmbH, Germany). This platform generates side-alternating vibrations. The manufacturer settings were set at a constant vibration frequency of 27 Hz. Actual frequency and peak-to-peak displacement were measured with an accelerometer (Actigraph GT9X Link) while loading the platform with an individual in standing and sitting posture. By locating the accelerometer at the exact position of the medial side of the midfoot (against vertical line number 1, which is six cm from the center axis and in the middle of the platform), actual parameters were calculated. The measurements confirmed sinusoidal vibrations and the results in x,y,z direction for each posture are shown in Table 2.

**Measurements.** The incongruent condition of the Stroop Color Word Interference Test (CWIT) was selected as neuropsychological test to examine the effects of WBV on attention and inhibition. CWIT is a measure of high reliability and validation, with the test being stable

**Table 1. Characteristics of the sample of healthy young adults (n = 60).**

| Characteristics | Total (n = 60) |
|---|---|
| General | |
| • Gender (M/F) | 17/43 |
| • Age (years) | 21.7 ± 1.96 |
| Physical domain | |
| • Height (cm) | 175.32 ± 8.96 |
| • Body mass (kg) | 70.18 ± 10.79 |
| • Body Mass Index (kg/m$^{-2}$) | 22.77 ± 2.55 |
| Medical history | |
| • Diagnosed with ADHD (n) | 3 |
| • Medication use of Ritalin (n) | 2 |
| Physical activity | |
| • Vigorous activity per week (minutes) | 174.25 ± 169.19 |
| • Moderate activities per week (minutes) | 250.93 ± 225.62 |
| Previous WBV experience (yes/no) | 10/48 |
| Caffeine consumption prior to the experiment (n) | Test 1: 6 |
| | Test 2: 1 |

**Table 2. Results of the actual frequency and peak-to-peak displacement.**

| Parameters | Sitting | Standing |
|---|---|---|
| Frequency (Hz) | • x: 27.1 | • x: 27.1 |
| | • y: 27.1 | • y: 27.1 |
| | • z: 27.1 | • z: 27.1 |
| Peak to Peak displacement (mm) | • x: 0.61 | • x: 0.57 |
| | • y: 0.09 | • y: 0.05 |
| | • z: 1.39 | • z: 1.39 |

*Note.* Frequency and peak-to-peak displacement was measured at the medial side of the midfoot while the vibrating platform was loaded with a participant for each posture.

over long periods of time and easy to administer and score [55]. The utilization of this test on selective attention and inhibition mimics previous studies [38, 40, 41].

Cards with 52 color names (blue, red, yellow, green) were presented on a screen. Each color-word name was presented in a different ink-color (e.g., the word red was colored blue). Twelve parallel versions of the CWIT were generated and one version was used as a practice version before the start of each session. The remaining versions were assessed after every experimental and control condition in a random order. Participants were asked to name the ink color of the 52 words correctly as fast as possible. Time in seconds needed to accomplish the test was measured with a stopwatch as outcome measure.

## Procedures

**Participant flow.** Fig 1 presents the flow of participants through each phase of the study. Sixty participants met the inclusion criteria and were randomly assigned to a starting condition (experimental or control) in a sitting or standing posture. After three experimental and control trials, CWIT was assessed. At least two days were in between sessions to minimize carry-over effects. In session two, the participants were subjected to the same procedures but started with the other starting condition and posture.

**General protocol.** Each participant had to come to the research lab twice. The protocol of both sessions was identical except for the posture applied: in one session a standing posture was used and in the other session a sitting posture. The sessions took place between 09:00h and 17:00h with a face-to-face presence of a test leader. Participants were asked to avoid caffeine consumption three hours prior to each experiment. To adhere to Dutch covid rules, participants agreed to wear a mouth-nose mask during each session. Prior to the experiment, two questionnaires were administered concerning general participant information and a short version of the International Physical Activity Questionnaire [56]. After completion of the questionnaires, participants were instructed to take off their shoes (socks only) and the general protocol was discussed. The protocol for each measurement in either a standing or sitting posture is shown in Fig 2. Participants were exposed to alternately three control and three WBV conditions with a duration of each two minutes. Directly after each condition, selective attention and inhibition was assessed with the incongruent condition of the Stroop CWIT. Each test was followed by a two-minute break. After completion of the six conditions, participants were asked to rate their perceived exertion, comfort and joy of the session.

**Postures.** A standing and sitting posture was used in the current study (Fig 3). In both postures, the feet (while wearing socks) were predominantly exposed to the vibrations by loading the midfoot on the platform. The medial sides of the feet were located against the vertical

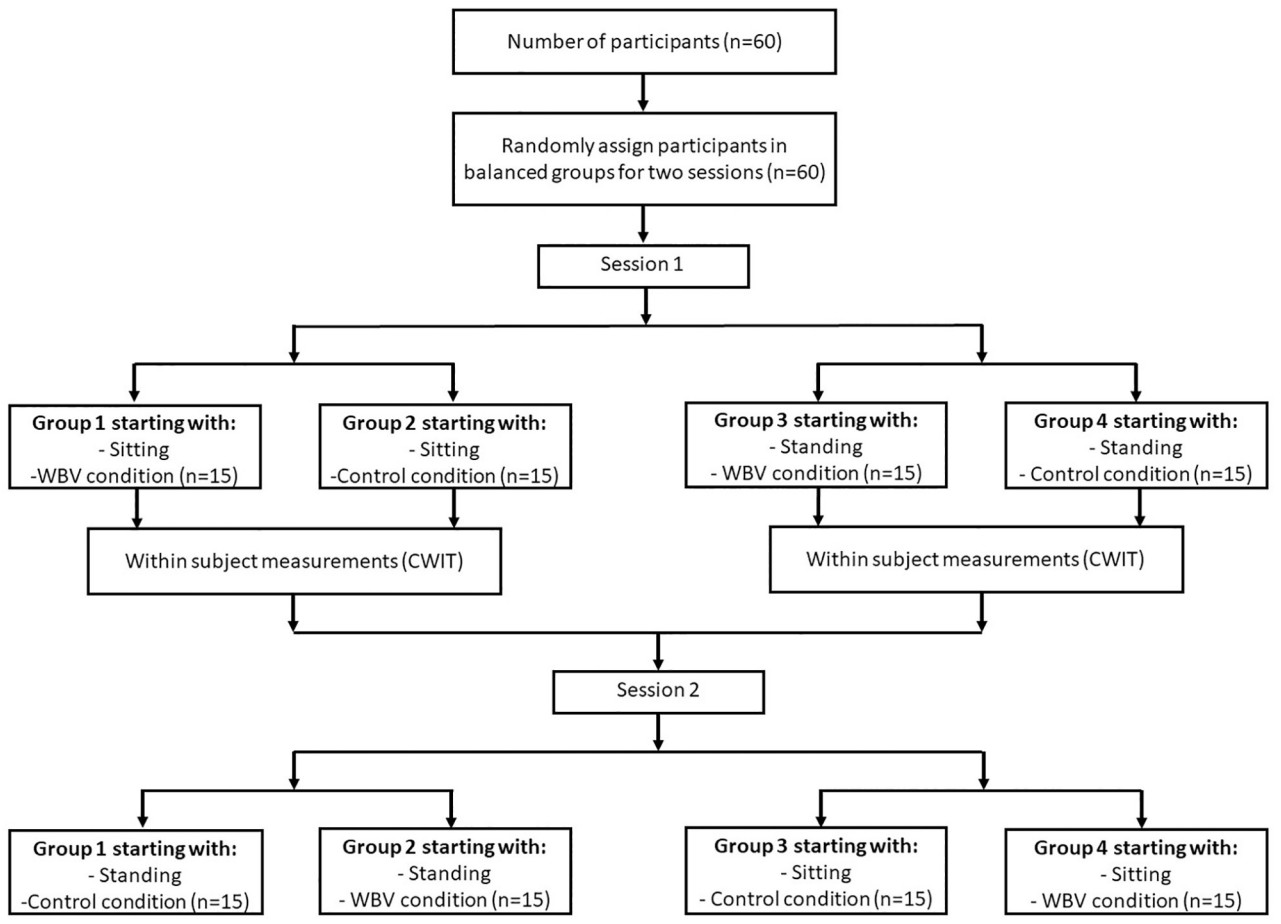

**Fig 1. Flow diagram of a balanced cross-over design assessing the effects of WBV versus control on cognition.** Participants were randomly assigned to four groups starting with different postures and conditions. Selective attention and inhibition were directly assessed after each trial. In session two, the other posture and starting condition was assigned to the participants following the same procedures.

line number 1 on the platform (Fig 4). In the standing posture, participants were instructed to stand on the platform while holding the handrail. Both skidding and vibration transmission to the head was prevented by slightly bending the knees in a squat position (Fig 3A). In the sitting posture, participants sat on a stool in front of the platform while resting their feet on the platform. Their hands were resting on both knees causing an indirect exposure to the vibrations. The stool was adjusted to each participant to maintain a knee flexion of approximately 90 degrees (Fig 3B). The control condition consisted of the exact same posture on, or in front of the platform without activating the WBV device.

## Statistical analysis

Normality of data distribution was confirmed after examining pre-processed data with visual inspection (plotting histograms) and shape of data (skewness, kurtosis). We performed a repeated measures ANOVA with respectively the scores of the CWIT as dependent variable and "Condition" (experimental, control) "Trial" (trial 1, trial 2, trial 3) and "Posture" (standing, sitting) as within-subjects factors. We calculated CWIT means of the three trials per condition by posture to examine significant interaction effects with paired t-tests (two-tailed). Finally, we used paired t-tests (two-tailed) to test differences in perceived comfort, joy and

## Session 1: Standing

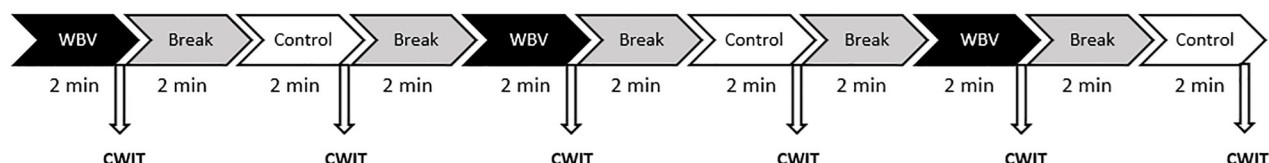

## Session 2: Sitting

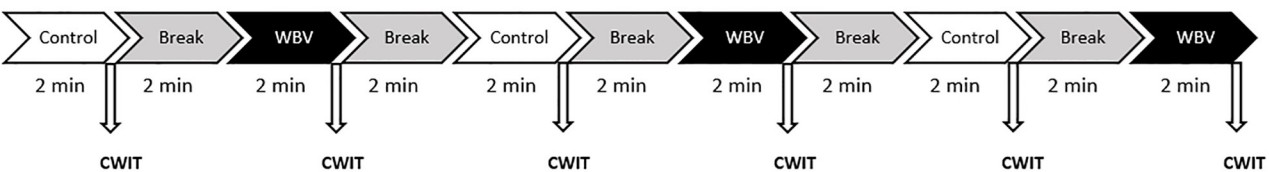

**Fig 2. Schematic example of the protocol of one participant.** Each participant underwent two sessions. The order of starting posture (standing, sitting) and condition (WBV, control) was balanced across subjects. Each condition took two minutes and after each condition the Color-Word Interference Test (CWIT) was assessed followed by a two minutes break.

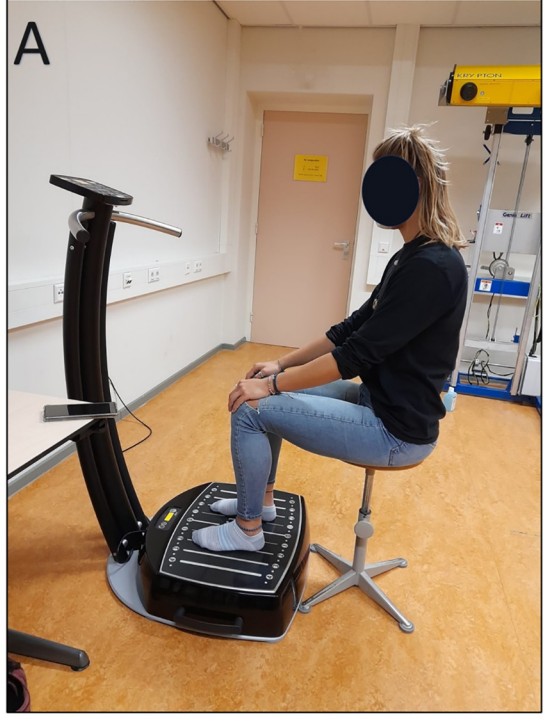
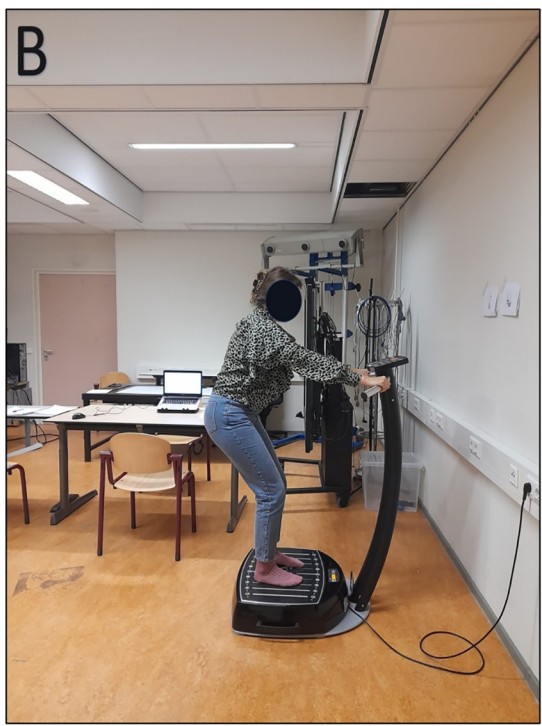

**Fig 3. The two postures used to examine the effects of WBV on cognition.** A: in the standing posture, participants held a static light squat while holding the handrail. B: in the sitting posture, participants were located on a stool in front of the platform with their knees bend in approximately 90 degrees while their hands were resting on the knees.

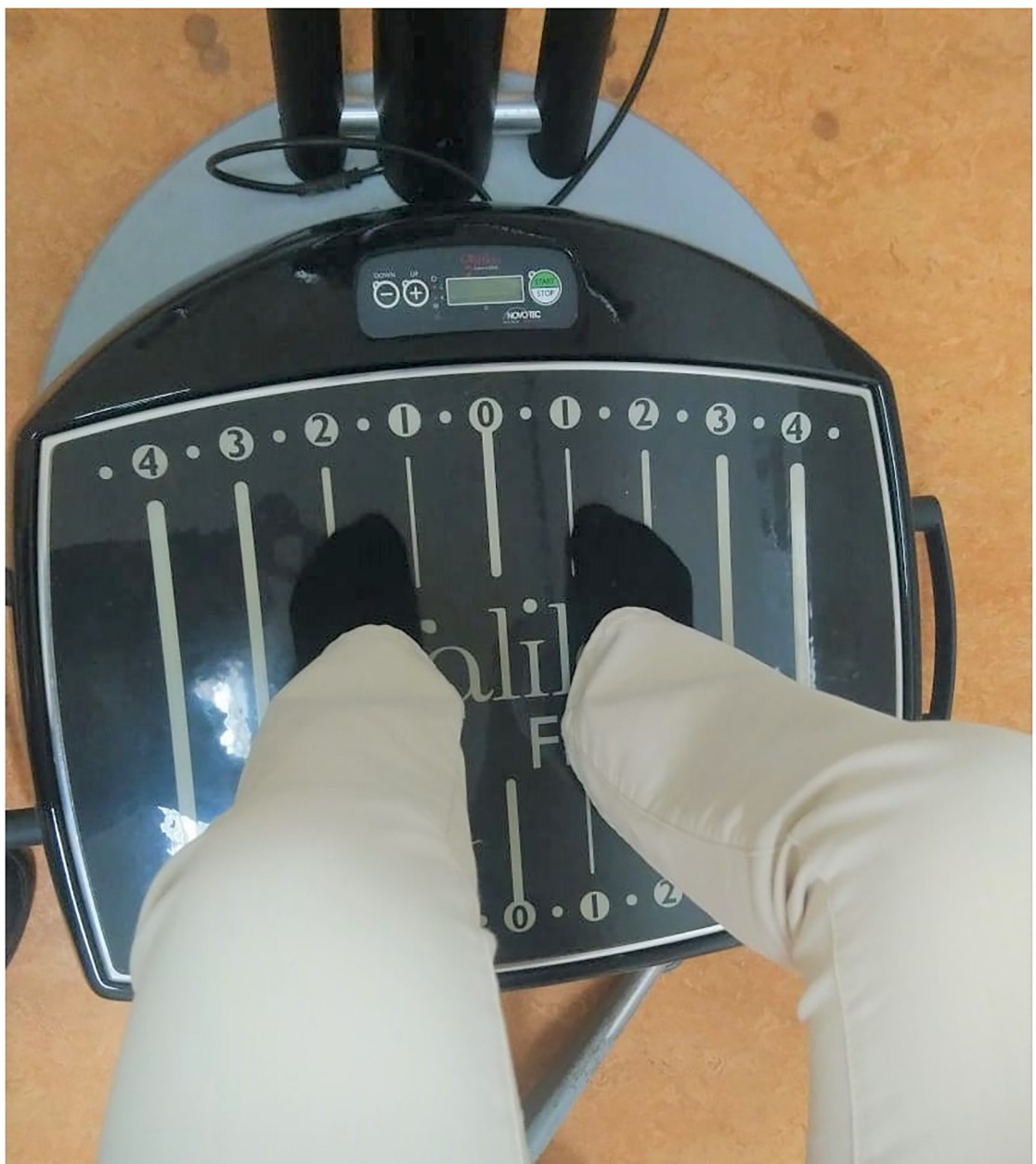

**Fig 4. Feet positioning on the Galileo platform in both postures.** The medial side of the feet were located in the middle of the platform against the vertical line number 1 which is approximately 6 cm from the center of the platform.

exertion between postures (sitting versus standing). Cohen's d was used as quantitative for effect size for paired t-tests and was calculated by dividing the mean difference by the standard deviation of the difference. A small effect corresponds to d = 0.2, a medium effect to d = 0.5 and a large effect to d = 0.8 [57]. We used $\eta^2$ effect sizes for ANOVA tests [57]. A small effect size corresponds to $\eta^2 = 0.01$, a medium effect size to $\eta^2 = 0.06$ and a large effect to $\eta^2 = 0.14$. Level of significance was set at $p < 5\%$ and SPSS version 26.0 was used for all analysis.

**Table 3. Results of the paired t-test comparing the performance on the Color-Word Interference Test (CWIT) between the WBV and control session in sitting and standing posture (n = 60).**

| Posture | WBV (Mean ± SD)[a] | Control (Mean ± SD)[a] | t | df | $p$[b] | Effect size (Cohen's D)[c] |
|---|---|---|---|---|---|---|
| Sitting | 33.42 ± 6.04 | 33.89 ± 6.52 | -2.29 | 59 | **0.026**[*] | 0.30 |
| Standing | 34.29 ± 6.00 | 34.05 ± 6.48 | 0.96 | 59 | 0.341 | -0.12 |

*Note*. Condition x Posture was significant (F(2,58) = 5.462, $p$ = 0.023, $\eta^2$ = 0.085).

[a] CWIT completion time in seconds; a lower completion time corresponds to better performance.

[b] two-tailed $p$-value.

[c] A positive effect size corresponds to an increase in performance after WBV vs. control, a negative effect size corresponds to a decrease in performance. Small effect: d = 0.20; Medium effect: d = 0.50; Large effect: d = 0.80.

[*] $p < 0.05$ indicates statistical significance.

## Results

### Effects of WBV on inhibition in standing and sitting posture

Table 3 shows the result of the CWIT directly after vibration and control in sitting and standing posture. All data can be found in S1 Dataset. The main effects of "Condition" (F(1,59) = 0.429, $p$ = 0.515, $\eta^2$ = 0.007) and "Posture" (F(1,59) = 1.010, $p$ = 0.319, $\eta^2$ = 0.017) were non-significant. However, the main effect of "Trial" (F(2,118) = 13.992, $p < 0.001$, $\eta^2$ = 0.192) appeared significant. One interaction effect was found for "Condition x "Posture" (F(2,58) = 5.462, $p$ = 0.023, $\eta^2$ = 0.085) indicating that posture influenced the effects of WBV on CWIT with a medium effect size. In sitting posture, performance on the CWIT improved on average 0.47 seconds (1.4% increase) after WBV as compared to control ($p$ = 0.026, d = 0.30). In standing posture, performance on the CWIT posture revealed a non-significant decrease of 0.22 seconds (0.7% decrease) ($p$ = 0.341, d = -0.12) (Table 3, Fig 5). The results did not change if participants with ADHD or participants using Ritalin were excluded.

### Perceived comfort, joy and exertion

On a scale of one to ten, WBV in sitting posture was perceived as "comfortable" (M = 7.8, SD = 1.8), "moderately fun" (M = 7.3, SD = 1.4) and "not exhausting" (M = 0.4, SD = 0.78). WBV in standing posture was perceived as "kind of comfortable" (M = 6.0, SD = 1.5), "moderately fun" (M = 6.9, SD = 1.4) and "relatively light exhausting" (M = 2.3, SD = 1.8). Sitting appeared more comfortable ($p < 0.001$), more joyous ($p$ = 0.014) and less exhaustive ($p < 0.001$) than standing (Fig 6).

## Discussion

The aim of this study was to investigate the short-term effects of side-alternating WBV on selective attention and inhibition in sitting and standing posture in young adults. Based on previous studies [38–42] with similar designs and supposed mechanisms, we hypothesized that side-alternating WBV would improve cognition in young adults. Indeed, exposing young adults to three bouts of two-minute side-alternating WBV (27 Hz) significantly improved the performance on the CWIT in sitting posture, but not in standing posture. The sitting posture was perceived as more comfortable, joyous and less exhaustive as compared to the standing posture.

The difference in effects in standing versus sitting posture indicates that posture moderates the effect of WBV on cognition. Presumably, potential underlying mechanisms affecting

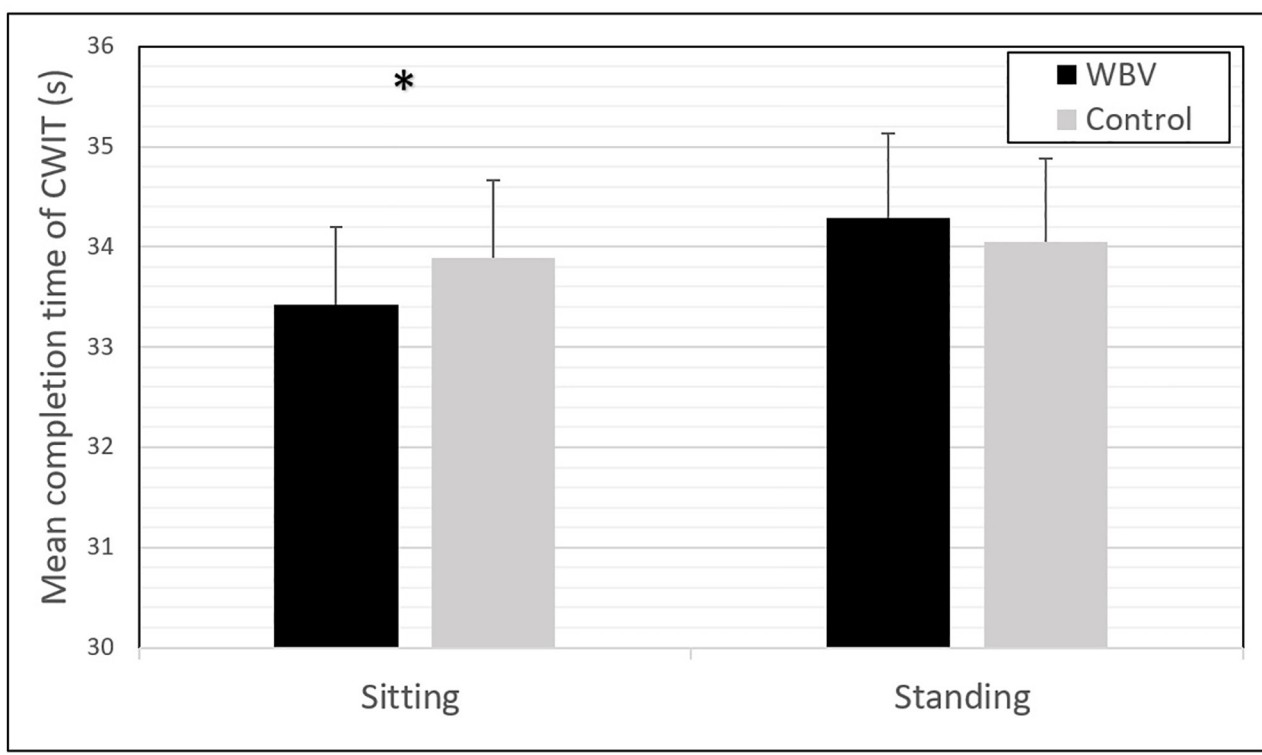

**Fig 5. Results of incongruent condition of Stroop Color-Word Interference Task after WBV and control.** Mean completion times and standard deviation of Stroop Color-Word Interference Test (CWIT) for WBV and Control conditions in standing and sitting posture. Lower mean completion times indicate less interference and a better inhibition. A significant positive effect after WBV was only found in the sitting posture. * $p < 0.05$ indicates statistical significance.

cognition are triggered in a different way resulting in different effects. One hypothesis states that vibration stimulates skin mechanoreceptors (Meissner corpuscles and Pacinian corpuscles) which affect neurotransmission in the prefrontal cortex and hippocampus leading to elevated cognition [24]. The density by which skin mechanoreceptors innervate the skin varies considerably across different body regions [58]. Fast adapting fibres (FA1 and FA2), innervating Meissner corpuscles and Pacinian corpuscles, are densely packed in the fingers and palmar surface of the hand. Compared with the hand, the foot sole is less densely innervated but also displays a relatively high innervation [58]. By exposing vibration to specifically these areas, a large number of skin mechanoreceptors are activated, possibly contributing to elevated cognition. In a sitting posture, cognition improved while the feet and palms of the hands were exposed to vibration. In the standing posture, cognition did not improve while only the feet were exposed to vibration. Thus, it seems that the sensory stimulation via skin mechanoreceptors is insufficiently triggered in the standing posture resulting in no effect. More research investigating underlying mechanisms of WBV affecting cognition is needed to confirm this.

Another hypothesis states that the constant initiation of stretch reflexes during WBV [34] might play a role in the cognitive improvements [24]. To our knowledge, the difference in muscle activation patterns in sitting versus standing posture during side-alternating WBV is not known. However, it is expected that the higher stretch of the muscle spindles and pre-activation of muscles in standing posture results in a larger increase in muscle activation as compared to the relaxed state of the muscles in the sitting posture [59–61]. Moreover, side-alternating WBV in standing posture has been extensively researched showing increased

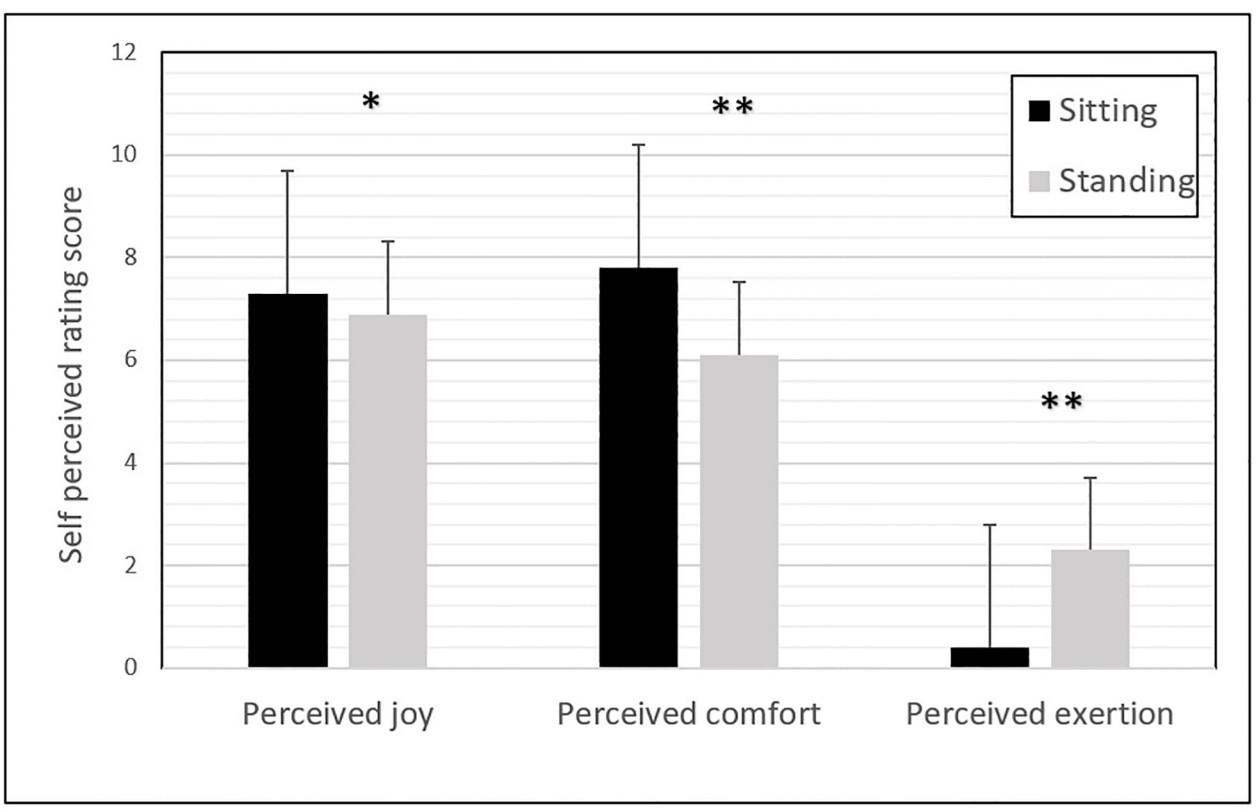

**Fig 6. Mean rating score of the perceived joy, comfort and exertion scale in sitting and standing posture.** The sitting posture was perceived as more joyous and comfortable than the standing posture. A higher perceived exertion was found in the standing postures as compared to the sitting posture. $^{*}$ $p < 0.05$. $^{**}$ $p < 0.001$.

muscle activity in neck, trunk [62], thigh [63] and lower limb muscles [64]. This suggests that increased muscle activity (stretch reflexes) might play a more dominant role as underlying mechanism in standing versus sitting posture. The absence of a superior effect after WBV versus control in standing posture might indicate that stretch reflexes do not positively affect cognition. However, this notion should be taken with caution. Although we did not find a positive effect in the present study, an fNIRS study investigating side-alternating WBV in standing posture showed increased oxygenation in the prefrontal and somatosensory cortex during WBV [25]. An increased prefrontal cortex oxygenation is closely linked to an increased performance on the Stroop task [65, 66]. This suggests that more research investigating cognition in relation to side-alternating WBV in standing posture is warranted.

The small positive effect of side-alternating WBV on cognition in sitting posture found in the present study is in accordance with prior studies investigating the effects of vertical WBV on cognition [38–42]. Since there is only little room for exercise-related improvements in these highly cognitive functioning and physically active individuals [67], both vertical and side-alternating WBV appears effective as cognitive-enhancing therapy in sitting posture.

As opposed to a 2% [41] and 5% [39] increase in performance on the CWIT after vertical WBV in healthy young adults, we only found an increase of 1.4% after side-alternating WBV. This may be explained by the posture used in the present study. Due to the tilting movement of the platform, it was not possible to mount a chair on the device. Therefore, participants had to sit in front of the WBV device with their feet on the platform and their hands on the knees.

Posterior body areas were therefore not in direct contact with a vibrating chair, which was the case in prior studies. Consequently, less afferent signals of cutaneous mechanoreceptors, which are found throughout the whole body [58], were transmitted to the sensory brain areas [23]. This may have reduced cognitive benefits in the present study. A future study investigating vertical versus side-alternating WBV in similar positions might bring more insights into the optimal type of vibration transmission affection cognition. The exact conjunction of potential underlying mechanisms affecting cognition remains unknown. However, exposing WBV to skin areas that are highly dense with cutaneous mechanoreceptors (palmar surface of the hand and feet) results in positive effects on cognition. More research into the underlying mechanisms affecting cognition is needed to confirm this statement. The present study further confirms that posture on the WBV device is an important factor contributing to the transmissibility of vibration through the human body [48]. This leads towards more insights in the search towards the optimal setting of WBV on cognition. Research investigating other settings (lower frequency, higher amplitude, other postures, higher temporal aspects or vertical versus side-alternating WBV) may bring more insights in the optimal setting of WBV on cognition. WBV further showed to be a comfortable, joyous and low-exhaustive intervention. This implicates that individuals who are compromised in physical and cognitive abilities can still benefit from the cognitive enhancing effects of WBV in sitting posture. The present study has a few limitations. First, the cognitive-enhancing effects of WBV were only investigated in healthy young adults. Generalizability of results to other populations is therefore limited. Future research should verify the cognitive-enhancing effects of WBV in populations with comprised cognition or physical ability. Second, the number of female and male participants was not balanced (17 male / 43 female). Third, only short-term cognitive effects of WBV were investigated. Research on chronic cognitive effects will elevate clinical relevance of WBV as cognitive-enhancing therapy. Fourth, only the incongruent condition of the Stroop CWIT was assessed in the present study, so an interference quotient between the CWIT and the Color-Block Test could not be calculated to isolate inhibitory control. However, previous studies investigating inhibitory control in healthy young adult revealed that the incongruent condition of the Stroop CWIT was most sensitive for WBV.

## Conclusion

In conclusion, the current study demonstrated that side-alternating WBV in sitting posture has a positive but small effect on cognition in healthy young adults with a high level of cognitive functioning. Side-alternating WBV in standing posture did not improve cognition indicating that posture on the WBV device may be a moderating for a cognitive response. Future studies should further explore the working mechanisms and optimal setting of side-alternating WBV affecting cognition, especially in individuals who are unable to perform active exercise.

## Supporting information

**S1 Dataset. Test scores of the Color-Word Interference Task for each WBV and control condition in standing and sitting posture.**
(DOCX)

## Acknowledgments

We thank Daniek van Laar, Corine Sijbesma, Jente Kerstholt and Marloes Wurkum for their assistance in the data collection.

## Author Contributions

**Conceptualization:** Y. Laurisa Arenales Arauz, Eddy A. van der Zee, Ype P. T. Kamsma, Marieke J. G. van Heuvelen.

**Formal analysis:** Y. Laurisa Arenales Arauz, Marieke J. G. van Heuvelen.

**Investigation:** Y. Laurisa Arenales Arauz, Marieke J. G. van Heuvelen.

**Methodology:** Y. Laurisa Arenales Arauz, Eddy A. van der Zee, Ype P. T. Kamsma, Marieke J. G. van Heuvelen.

**Project administration:** Y. Laurisa Arenales Arauz, Marieke J. G. van Heuvelen.

**Validation:** Y. Laurisa Arenales Arauz, Marieke J. G. van Heuvelen.

**Visualization:** Y. Laurisa Arenales Arauz, Eddy A. van der Zee, Ype P. T. Kamsma, Marieke J. G. van Heuvelen.

**Writing – original draft:** Y. Laurisa Arenales Arauz, Eddy A. van der Zee, Ype P. T. Kamsma, Marieke J. G. van Heuvelen.

**Writing – review & editing:** Y. Laurisa Arenales Arauz, Marieke J. G. van Heuvelen.

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
