## [Decision Letter · Decision Letter 0]

16 Nov 2022

PONE-D-22-16452Short-term effects of side-alternating Whole-Body Vibration on cognitive function of young adultsPLOS ONE

Dear Dr. van Heuvelen,

Thank you for submitting your manuscript to PLOS ONE. After careful consideration, we feel that it has merit but does not fully meet PLOS ONE’s publication criteria as it currently stands. Therefore, we invite you to submit a revised version of the manuscript that addresses the points raised during the review process. Based on the reviewer comments and suggestions, it is required to provide a revision of the manuscript.

We look forward to receiving your revised manuscript.

Kind regards,

Giancarlo Condello, Ph.D.

Academic Editor

PLOS ONE

https://journals.plos.org/plosone/s/fileid=ba62/PLOSOne_formatting_sample_title_authors_affiliations.pdf.

Reviewers' comments:

Reviewer's Responses to Questions

**Comments to the Author**

1. Is the manuscript technically sound, and do the data support the conclusions?

Reviewer #1: Yes

Reviewer #2: Yes

2. Has the statistical analysis been performed appropriately and rigorously? 

Reviewer #1: Yes

Reviewer #2: Yes

3. Have the authors made all data underlying the findings in their manuscript fully available?

Reviewer #1: Yes

Reviewer #2: Yes

4. Is the manuscript presented in an intelligible fashion and written in standard English?

Reviewer #1: Yes

Reviewer #2: Yes

5. Review Comments to the Author

Reviewer #1: The subject of this study is highly important. The scientific interest in the clinical application of whole-body vibration (WBV) is increasing in the world. Different biological responses to the WBV have been described. It is relevant to know about the effect of the WBV on cognition, and this study using a side-alternating vibrating platform shows important findings. It is presenting a proper design of the study. I am presenting some suggestions to aid in the improvement of the format of the manuscript, that are in the attached fike.

Reviewer #2: PONE-D-22-16452

Short-term effects of side-alternating Whole-Body Vibration on cognitive function of young adults

PLOS ONE

The objective of this research was conducted to investigate the short-term effects of side alternating WBV on selective attention and inhibition in sitting and standing posture in young adults.

The approach is original. The manuscript reads smoothly and is easy to understand. The aims, scope, and results of the study are clearly stated. I have very much enjoyed reading this paper. I find it interesting and clearly written and satisfying also all the other publication criteria of the “PLOS ONE”. The study provides a very valuable addition to this line of research, and adds relevantly to the subject with additional original findings. I thus find that this paper definitively delivers results that will surely be of interest to the readership of the “PLOS ONE”. I recommend the publication of this interesting paper.

6. PLOS authors have the option to publish the peer review history of their article (what does this mean?). If published, this will include your full peer review and any attached files.

Reviewer #1: **Yes: **Mario Bernardo-Filho

Reviewer #2: No

---

## [Author Response · Author response to Decision Letter 0]

13 Dec 2022

We responded to all comments of editor and reviewers in the response to reviewer document. We have no other specific comments.

---

## [Editor Report · Decision Letter 1]

20 Dec 2022

Short-term effects of side-alternating Whole-Body Vibration on cognitive function of young adults

PONE-D-22-16452R1

Dear Dr. van Heuvelen,

We’re pleased to inform you that your manuscript has been judged scientifically suitable for publication and will be formally accepted for publication once it meets all outstanding technical requirements.

Kind regards,

Giancarlo Condello, Ph.D.

Academic Editor

PLOS ONE
---

## [Editor Report · Acceptance letter]

3 Jan 2023

PONE-D-22-16452R1 

Short-term effects of side-alternating Whole-Body Vibration on cognitive function of young adults 

Dear Dr. van Heuvelen:

I'm pleased to inform you that your manuscript has been deemed suitable for publication in PLOS ONE. Congratulations! Your manuscript is now with our production department. 

Kind regards, 

on behalf of

Dr. Giancarlo Condello 

Academic Editor

PLOS ONE